# Low-Light Sparse Polarization Demosaicing Network (LLSPD-Net): Polarization Image Demosaicing Based on Stokes Vector Completion in Low-Light Environment

**DOI:** 10.3390/s24113299

**Published:** 2024-05-22

**Authors:** Guangqiu Chen, Youfei Hao, Jin Duan, Ju Liu, Linfeng Jia, Jingyuan Song

**Affiliations:** 1Electronics and Information Engineering Institute, Changchun University of Science and Technology, Changchun 130022, China; guangqiu_chen@126.com (G.C.); youfei_hao@163.com (Y.H.); liuju_2020@163.com (J.L.); linfeng_2023@163.com (L.J.);; 2Space Opto-Electronics Technology Institute, Changchun University of Science and Technology, Changchun 130022, China

**Keywords:** polarization image demosaicing, low light, Stokes vector completion, hourglass network

## Abstract

Polarization imaging has achieved a wide range of applications in military and civilian fields such as camouflage detection and autonomous driving. However, when the imaging environment involves a low-light condition, the number of photons is low and the photon transmittance of the conventional Division-of-Focal-Plane (DoFP) structure is small. Therefore, the traditional demosaicing methods are often used to deal with the serious noise and distortion generated by polarization demosaicing in low-light environment. Based on the aforementioned issues, this paper proposes a model called Low-Light Sparse Polarization Demosaicing Network (LLSPD-Net) for simulating a sparse polarization sensor acquisition of polarization images in low-light environments. The model consists of two parts: an intensity image enhancement network and a Stokes vector complementation network. In this work, the intensity image enhancement network is used to enhance low-light images and obtain high-quality RGB images, while the Stokes vector is used to complement the network. We discard the traditional idea of polarization intensity image interpolation and instead design a polarization demosaicing method with Stokes vector complementation. By using the enhanced intensity image as a guide, the completion of the Stokes vector is achieved. In addition, to train our network, we collected a dataset of paired color polarization images that includes both low-light and regular-light conditions. A comparison with state-of-the-art methods on both self-constructed and publicly available datasets reveals that our model outperforms traditional low-light image enhancement demosaicing methods in both qualitative and quantitative experiments.

## 1. Introduction

Polarization is an outward manifestation of the transverse wave nature of light and carries information about the structure, material, and reflection angle of the target surface, which can complement the texture information that is passively obtained from the scene. Many polarization imaging techniques have been devised and polarization information is used in a wide range of military and civilian applications, such as in photometrically challenging object attitude estimation [1], the segmentation identification of objects with special materials (metallic, highly exposed or transparent) [2], reflection removal [3], polarization information color constancy [4], and material classification [5].

Traditional DoFP polarimetric imaging devices possess significant advantages in real-time imaging, and thus are widely employed at present. The structure of the polarizer and color filter distribution form of its sensor is shown in Figure 1a, and the existing color polarization imaging instruments are shown in Figure 1b. Although the real-time performance of this structure can be guaranteed, the Bayer structure + polarizing filter structure degrades the quality of the RGB image, especially in low-light environment with less photon transmission. As a result, polarized imaging produces severe noise and artifacts, in addition to a degradation in the quality of the RGB image. For advanced applications of polarized images (e.g., semantic segmentation and target detection), the effect of noise makes the model complex and difficult to converge. It is important to emphasize here that the source of the noise is mostly Gaussian noise, which is further amplified in the subsequent processing and, hence, is reflected in the DoLP and AoP images [6]. Although increasing the ISO of the camera can make the image brighter, the ensuing noise and chromatic aberration are unavoidable, and such noise and distortion have a more serious impact on the subsequent resolution of polarized images. Additionally, increasing the exposure time will inevitably reduce the frame rate, which can also lead to blurring and artifacts [7].

In recent years, there have been many studies related to low-light image enhancement; however, there are currently no studies that can guarantee low-light image enhancement while realizing a distortion-free resolution of polarization information for special polarization sensor structures. The most straightforward approach to enhance low-light images is to linearly amplify each pixel. A typical example is shown in Figure 2, where it can be seen that although the method results in an increase in S0 brightness, at the same time, the noise is amplified. Moreover, due to the amplification of the intensity of image noise, the polarization image (DoLP, AoP) noise, which is then solved by the nonlinear operation, is even more amplified. Figure 3 shows an example of the effect of noise; in the first row, we simulate a non-polarized environment, i.e., the pixel value for acquiring the DoLP is 0. In the second line, we simulate the introduction of a small amount of Gaussian noise (σ=0.04) into the camera component, and it can be seen that the DoLP at this point has a large value of DoLP in the regions of the image with low intensity. This shows that a weak noise can present a large number of errors in DoLP images. Therefore, it is essential for our method to consider the effect of noise.

For the enhancement of low-light images, there are currently two main types of methods, i.e., traditional methods, including non-local block matching methods [8] and hypothesis-based methods [9], and deep learning methods. The research papers [10,11] adopted the deep learning approach. Although the above traditional or deep learning-based low-light image enhancement methods excel in image denoising and bias color correction, these methods do not take into account the peculiarities of low-light polarized images; thus, these methods are not applicable to low-light polarized image enhancement tasks. The reason is as follows: Stokes vector is obtained for the sum or difference operation of intensity images in different directions, and none of the traditional methods in the process of intensity image enhancement consider the light intensity change characteristics between polarized intensity images in different directions, i.e., the correlation of the intensity values of the corresponding pixel points of the images with different polarized intensities. This causes the resolved polarized image to be distorted and produces a lot of noise and artifacts. In addition, in the polarization image solving task, the traditional polarization demosaicing methods [6,12,13] also only solve the polarization image by the enhanced polarization intensity image, and the interpolation process still results in a large amount of noise; moreover, all of them fail to take into account the quality degradation of the RGB image due to the polarization filters.

Based on the above, this paper designs a multi-task convolutional neural network structure, LLSPD-Net, in terms of both low-light polarization intensity image enhancement and low-noise high-quality imaging of polarized images. The structure fully takes into account the correlation between image RGB channels and polarization intensity images at 0°, 45°, 90°, and 135°, as well as the advantage of Stokes vector complementation, which solves the problems of low-light polarization image enhancement and high-quality polarization simultaneous imaging.

In summary, this paper addresses two key issues. The first is the problem of intensity image enhancement in a low-light environment. The second is the problem of noise generated by polarization imaging in a low-light environment with serious artifacts. The main contributions of this paper can be summarized in the following threeaspects:(1)An imaging model for polarized images in low-light environments, LLSPD-Net, is designed to generate polarized images with less noise while enhancing low-light polarized intensity images.(2)In order to obtain high-quality RGB images and polarized images at the same time, we designed a Stokes complementary method to acquire polarized images with the help of the hourglass network structure, and simulated the sparse arrangement of polarization filters.(3)We collected a low-light polarization dataset, L-Polarization, containing different materials for indoor, outdoor, and different scenes, which is a paired dataset containing 300 sets of low-light and constant-light environments.

## 2. Related Works

### 2.1. Low-Light Image Enhancement Methods

For low-light image enhancement methods, related research can be broadly categorized into two main groups: traditional methods as well as deep learning-based methods. These two main categories of methods in low-light enhancement at the same time are inevitably considered the work of image denoising, and denoising is essential. One of the more typical traditional methods is the C-BM3D [14], which utilizes highly sparse local 3D transform domains in each channel of the luminance–chrominance color space for the design of filtering to achieve the denoising of low-light images and visual quality enhancement. With the development of deep learning in recent years, Jiang et al. [15] used an unsupervised generative adversarial network architecture to solve the problem of low-light image enhancement in the absence of low-light–normal-light image pairs. Li et al. [16] designed a trainable convolutional neural network (CNN) structure for microlight image enhancement to obtain enhanced images using the Retinex model. Ma et al. [17] established a cascading illumination learning process with weight sharing for low-light image enhancement and defined the associated unsupervised training loss to improve the generalization of their model. The above low-light processing models have achieved good results in image denoising and color correction, but they are not satisfactory for low-light polarized images. Since the polarization information reflects the difference between the intensity images [18], the enhancement process without considering the correlation between the four polarization directions of intensity images—$I_{0},I_{45},I_{90},I_{135}$—will result in a low quality of the computed color S0 as well as problems such as DoLP and AoP noises, distortions, and severe artifacts.

### 2.2. Polarization Demosaicing and Depth Completion Methods

Current demosaicing methods for DoFP sensors can be broadly categorized into three groups, namely sparse representation-based, interpolation-based, and end-to-end deep learning-based methods. Typical studies of sparse representations include the research of Luo et al. [19], who combined the demosaicing task into an optimization problem to be solved by introducing RGB polarization channel correlation, adaptive subdictionary, and non-local self-similarity constraints. In addition, traditional methods broadly use interpolation, for example, Li et al. [20] proposed a technique for polarization demosaicing using Newtonian polynomial interpolation, which takes into account the interpolation error and chooses to interpolate in the polarization difference domain. For deep learning, Zeng et al. [21] designed an end-to-end fully convolutional neural network architecture, which aims to directly improve the imaging quality of the three polarization characteristics of intensity (S0), degree of line polarization (DoLP), and angle of polarization (AoP), and does not interpolate the polarization intensity images.

All of the above demosaicing methods for DoFP sensors have achieved some significant results, but none of them considered the problem of RGB image quality degradation due to the polarization filter, i.e., the polarization filter reduces the photon transmittance problem. In this paper, a sparsity DoFP sensor polarization demosaicing method based on Stokes vector complementation is designed in conjunction with previous studies, taking into account the polarization imaging in low-light conditions and the structural properties of DoFP sensors. We were inspired mainly by the idea of deep complements, such as the work of Zhang et al. [22]. Our method takes an RGB image as input and predicts dense surface normals and occlusion boundaries to solve the problem of missing pixels in the original observation. Hegde et al. [23] utilized an exact sparse depth as input to the RGB image to generate a dense depth map. The method focuses on a quadtree decomposition modeling approach. However, all of the above complementation methods are designed to be specific and, therefore, cannot be directly applied to the task of polarization information complementation.

## 3. Method

### 3.1. Overview

The aim of this paper is to enhance low-light images while ensuring that the polarization information is accurate and not true imaging. The calculation of the polarization Stokes vector is shown by Equation (Equation 1) [24], which is closely related to the enhanced polarized intensity image. Therefore, it is our aim to try to avoid the loss of polarization information while ensuring the enhancement of low-light images. Our second aim concerns the fact that DoFP sensors have a low photon transmittance due to their own structure, and polarization demodulation often requires a process of “sampling + interpolation + computation”. This results in severe noise and distortion, so we compensate for missing pixels in the Stokes vector map directly through a deep learning network with the help of a sparse sensor structure proposed in the literature [25]. This method is able to ensure high-quality RGB images while obtaining polarized images with less noise.
(1)$$\begin{array}{c} \left\{\begin{array}{c} S_{0}=\frac{1}{2}\left(I_{0}+I_{45}+I_{90}+I_{135}\right)\\ S_{1}=I_{0}-I_{90}\\ S_{2}=I_{45}-I_{135} \end{array}\right. \end{array}$$

To solve the above two problems, we constructed a multi-task convolutional neural network architecture: LLSPD-Net. As shown in Figure 4, our entire network architecture consists of two sub-networks, the intensity network and the polarization complementation network (PCN), where the input of the PCN is the output of the intensity network after enhancing the low-light image; the specific structure of its two-part network is described in detail in Section 3.2.1 and Section 3.2.2.

The raw input to our network is a low-light DoFP sensor RAW image, which is decomposed by data to obtain an RGB image and four polarization direction intensity images. The intensity image is then fed into the intensity network for low-light enhancement. Finally, the enhanced RGB image is used as the high-resolution image input to guide the compensation of the Stokes vector using PCN. The enhanced $I_{0}$, $I_{45}$, $I_{90}$, and $I_{135}$ are used to generate sparse Stokes vectors, which are then predicted by the PCN network for the missing pixels of the sparse Stokes components to obtain the high-resolution Stokes vectors. S0 is acquired by utilizing the gain absorption sensitivity difference for the enhanced RGB image and thus obtaining the high-resolution S0 component.

### 3.2. Network Architecture

#### 3.2.1. Intensity Network

As shown in the first branch of Figure 4 for the intensity network, we feed the mosaiced RGB image, as well as the polarized light intensity images in the four polarization directions of 0°, 45°, 90°, and 135° into this network to enhance the intensity image. Inspired by the literature [26], our network architecture consists of three main components. First, the low-light image is decomposed into an illuminance map and a reflectance map by three convolutional layers; then, the decomposed image is used as input, and the noise in the reflectance map is suppressed by using the illuminance map as a constraint through six convolutional layers to realize the denoising purpose. Finally, the illuminance map obtained in the first part along with the reflectance map obtained in the second part are fed to the DRM module, through which the detail reconstruction module is finally used to obtain the ortho-light image with better visual quality. Due to the special jump connection structure of residual networks, it can make deep neural networks easier to optimize in the training phase without causing gradient vanishing or explosion. Therefore, inspired by the literature [27], we used multiple residual modules (RMs) in the network to obtain better decomposition and denoising results. The specific structure of the RM module is shown in Figure 5, where each RM contains five convolutional layers with convolutional kernel sizes of 1, 3, 3, 3, and 1, respectively. We added a 64 × 1 × 1 convolutional layer at the shortcut connection. Each RM has a 64 × 3 × 3 convolutional layer before and after it. Due to some of the correlation between our four polarized intensity images and the RGB images, we chose to process the correlation channels jointly. In addition, our ultimate goal was to achieve high-quality polarization imaging, so we added the intensity network-enhanced RGB images and the four polarization intensity images into the polarization information complementary network (PCN), which further ensured high-quality, high-fidelity imaging of the polarization images by means of Stokes vectors.

#### 3.2.2. Polarization Completion Network

As shown in Figure 6, our PCN network is a Stokes vector complementary network, and the overall structure contains two hourglass network branches, i.e., the RGB image guidance branch and the polarization Stokes vector complementary branch. The inputs of the network are the high-quality RGB images output from the intensity network and the sparse Stokes components S_1,2_ after enhancement, and the output of the network is the complementary full-resolution dense S_1,2_.

The RGB image is fed into the RGB image guidance branch as a full-resolution high-quality guidance image. The image is first fed into the hourglass network by a 5 × 5 convolutional coding, and the first hourglass structure is used to extract the RGB image information. The aim is to provide guidance for the second stage of the densification complement of polarized Stokes, whereby the second stage is the Stokes completion branch; this stage densifies the input sparse Stokes vector. The overall network consists of two hourglass units similar to the symmetric structure of Unet, and we also introduced two modules, channel attention block (CAB) [28], and repetitive guidance module (RG) [29], into the network. Among them, the CAB module is used to extract global features and visual features in complex scenes, using encoder-side important feature weighting, which is added with decoder features to achieve a more accurate information guidance for RGB branches.

The RG module has the function of repetitive guidance, which enhances the complementation of S_1,2_ information through gradual refinement. The specific structure is shown on the right side of Figure 6, which takes $D_{\mathrm{ij}}$ and $e_{1j}$ as inputs, and finally obtains the refined $d_{\mathrm{jk}}$ through the gradual refinement process. Then, the final polarization $d_{j}$ is obtained by adaptive fusion. The RG structure mainly consists of an efficient guidance (EG) module and an adaptive fusion (AF) module [29], where EG first stitches RGB images and Stokes vector maps together by 3 × 3 convolution, and then generates C × 1 × 1 features by global average pooling. Finally, a point-by-point operation is performed between the features and the Stokes input. The AF module is the union of many coarse Stokes features ($d_{j1},---,d_{\mathrm{jk}}$) obtained by repeated guidance to generate refined Stokes vector maps. This structure introduces an adaptive fusion mechanism to refine the Stokes vector map.

In Figure 6, $E_{ij}$ in the encoder takes $E_{i(j-1)}$ and $D_{(i-1)j}$ as inputs. In the decoder, $D_{ij}$ takes $E_{ij}$ and $D_{i(j+1)}$ as inputs. The process when i>1 is as follows:(2)$$\begin{array}{cc} E_{ij} & =\left\{\begin{array}{cc} \mathrm{Conv}\left(D_{(i-1)j}\right), & j=1,\\ \mathrm{Conv}\left(E_{i(j-1)}\right)+D_{(i-1)j}, & 1<j\leq 5, \end{array}\right.\\ D_{ij} & =\left\{\begin{array}{cc} \mathrm{Conv}\left(E_{i5}\right), & j=5,\\ \mathrm{Deconv}\left(D_{i(j+1)}\right)+E_{ij}, & 1\leq j<5, \end{array}\right. \end{array}$$
In the above equation, Deconv (·) is the inverse convolution operation and $E_{1j}$$=\mathrm{Conv}\left(E_{1(j-1)}\right)$.

### 3.3. Loss Function

Our loss is divided into two parts, the first part is the intensity loss that constrains the image enhancement of low-light intensity, and the second part is the Stokes loss that realizes the Stokes vector complement. Intensity loss mainly consists of two parts: content loss and perception loss. For Stokes loss, we mainly use the mean-squared error (MSE) to calculate the loss.

Specific definitions are given below:(1)In the intensity loss, we use $\ell_{1}$ loss as the content loss, and compute the perceptual loss using the features extracted from the VGG-16 pre-trained model before the activation layer. The specific definitions are as follows:
(3)$$L_{\mathrm{con}\;}=\frac{1}{N}\sum_{i=1}^{N}\left|{\hat{G}}_{\mathrm{low}\;}^{i}-G_{\mathrm{gt}}^{i}\right|$$
(4)$$L_{\mathrm{per}\;}=\frac{1}{N}\sum_{i=1}^{N}\frac{1}{C_{j}H_{j}W_{j}}∥\phi_{j}\left({\hat{G}}_{\mathrm{low}\;}^{i}\right)-\phi_{j}\left(G_{\mathrm{gt}\;}^{i}\right)∥$$
where i∈[1,N] denotes the *i*th sample used to compute the loss and *j* denotes the *j*th layer of the VGG-16 pre-trained model. That is, the strength loss expression is as follows:
(5)$$L_{G}\left(\hat{G}\right)=L_{\mathrm{con}}+\lambda_{1}L_{\mathrm{per}\;}$$(2)Stokes loss is defined as
(6)$$L_{S}\left(\hat{S}\right)={∥{\hat{S}}_{1,2}-S_{1,2}^{\mathrm{gt}}∥}_{2}$$
(7)$$L_{\mathrm{Stokes}\;}=L_{S}\left(\hat{S}\right)+\lambda_{2}\left\{L_{S}\left({\hat{S}}^{1\mathrm{st}}\right)+L_{S}\left({\hat{S}}^{2\mathrm{nd}}\right)\right\}$$
where $s_{1,2}^{\mathrm{gt}}$ is the ground truth.

In summary, the overall loss function formula is as follows:(8)$$f=\arg\min\left[\lambda_{3}L_{\mathrm{Stokes}\;}+L_{G}\left(\hat{G}\right)\right]$$

In Equation, $\lambda_{1}$ is used as a balance factor between content loss and perceptual loss, and its optimal value was empirically determined to be 0.1. $\lambda_{2}$ is a hyperparameter that decreases with the number of epochs, and the initial value was empirically set to 0.2. And $\lambda_{3}$ is a balance parameter between Stokes loss and intensity loss, which was empirically set to 0.6.

## 4. Experimental Section

### 4.1. Experimental Configurations

(1)Dataset:

In this paper, the L-Polarization dataset is acquired using LUCID PHX050S-QC camera, manufactured by LUCCID Canada, and used to train, validate, and test our LLSPD-Net. The dataset takes into full consideration the large polarization difference between metal and dielectric materials, etc., and collects indoor and outdoor scene data with different shapes and materials, such as metal, plastic, wood, and fabric. Among them, the sparse polarization images used in this paper are acquired from reference [25]. In order to more clearly reflect the diversity of the training data in the dataset of this paper, we crop the images of the dataset into 64 × 64 subimages and map these subimages to a point within the unit circle. As shown in Figure 7, we compare the polarization dataset under two low-light conditions: L-Polarization and LLCP [30]. The x-axis and y-axis of the graph are defined as the median pixel values of $s_{1}/s_{0}$ and $s_{2}/s_{0}$, respectively. Each subimage is then grouped into one of ten classes. We classify them according to the magnitude of the $a={\left(s_{1}/s_{0}\right)}^{2}+{\left(s_{2}/s_{0}\right)}^{2}$ values, where we classify the values of $a<=P_{L}^{2}$ as low-polarization subimages, and those of $a>=P_{U}^{2}$ as high-polarization subimages, where $P_{L}$ and $P_{U}$ denote the threshold boundary values for the first and the tenth categories, respectively, and the rest of the eight categories are identified by dividing the unit element centroid ring region into equal areas, which correspond to the different regions of the AoP. The $P_{L}$ and $P_{U}$ in the first column of Figure 7 are set to 0.1 and 0.4, respectively, and the distribution of the data can be seen through the point mapping of the two datasets. The vertical axis in the second column indicates the median intensity value of S0, while the vertical coordinate in the third column is the median local binary pattern (LBP) [31] of S0, reflecting the texture diversity of the subimage.

As can be seen from the figure, our L-Polarization contains a large number of highly polarized subimages (nearly 2%), reflecting more polarization diversity. Moreover, the L-Polarization dataset exhibits greater texture diversity in terms of median LBP compared to the LLCP data.

(2)Training details:

We constructed training, testing, and validation datasets from 300 sets of low-light data in an 8:1:1 ratio. Our network architecture was implemented on a PC equipped with an NVIDIA A100 GPU. The epoch for the whole model training was set to 50, the batch size was 3, the initial learning rate was 0.003, and for the adaptive optimizer, we chose the Adam optimizer with better noise immunity.

### 4.2. Experimental Results

#### 4.2.1. Comparison with Low-Light Image Enhancement Methods

We chose three mainstream low-light enhancement models to compare with our method. C-BM3D [14] is the traditional method, EnlightenGAN [15] and LightenNet [16] are the deep learning methods. We first compared the enhanced RGB images in terms of color, contrast, and brightness, and we also compared the enhanced RGB and polarization results with their respective ground truth (GT). Figure 8 shows the qualitative result analysis of the enhanced RGB image, the complementary Stokes vectors S1 and S2, and the generated DoLP and AoP images, which show that our model is the closest to the ground truth, and that our model works well for the polarization imaging of different materials with large polarization differences. The most positive thing is that our LLSPD-Net suppresses the obvious noise of DoLP and AoP images while guaranteeing the robust characterization of the edge information of polarized images. As shown in the figure, C-BM3D has a poor brightness effect for low-light image enhancement, but it has a strong ability to maintain the polarization state of polarization intensity images in different directions. Unfortunately, the method introduces a large amount of noise, which makes the generated DoLP and AoP produce more serious noise. Although EnlightenGAN enhances the brightness, color, and other information of the RGB image, it produces obvious distortion compared to the ground truth. Due to the existence of this distortion, it will have a great impact on the subsequent compensation of the Stokes vector and the acquisition of DoLP and AoP; furthermore, it can be seen that the polarization imaging results of EnlightenGAN noise, artifacts, and distortion phenomena are very obvious. For the LightenNet method, it is not ideal in terms of RGB image enhancement, but the method is better in terms of polarization state preservation; and although the polarization imaging effect is better, it is still far from the ground truth. In conclusion, it can be seen that our method is closest to the ground truth, and the enhanced RGB image is the best in terms of visual effect and polarization state preservation.

Qualitative comparison: We selected 20 sets of images from each dataset, the public dataset LLCP and the self-constructed dataset L-Polarization, for quantitative testing. We chose four quantitative metrics such as structural similarity (SSIM), peak signal-to-noise ratio (PSNR), patch-based contrast quality index (PCQI) [32], and angle error [33], which is a measure of the angle error of the polarization angle image to judge the performance of low-light image enhancement and polarization imaging. Among them, SSIM mainly evaluates the contrast and structural fidelity of the generated results, and a higher value indicates a higher image quality. PSNR measures the distortion of polarization and RGB images, and a higher value indicates that the image is richer in information and is less noisy. PCQI measures the contrast information of the results, and a higher value indicates that the contrast of the image is better. Angle error denotes the generating error of the AoP image. The smaller the angular error, the better the performance of the model. The average metrics of the test results are shown in Table 1, and the test results show that LLSPD-Net reflects the optimal effect in several metrics, proving that the method of this paper has a good information fidelity advantage in terms of RGB image enhancement in low-light environments and polarization image generation, and is able to maximize the high-contrast, low-noise imaging of polarization images while ensuring the enhancement of RGB images.

#### 4.2.2. Comparison with the Basic Polarization Demosaicing Method

In order to reduce the noise of polarization imaging, we discard the traditional interpolation idea, but draw on the idea of deep completion, and use the end-to-end mapping of deep learning models to directly complement the Stokes vector for polarization imaging. Therefore, we use the LLSPD-Net Part I intensity network-enhanced intensity images with different polarization directions as source images, and compare them with several existing typical polarization demosaicing (PDM) models. Among them, Newton’s [20] and bicubic [34] interpolations are traditional methods, while ForkNet [21] is a deep learning method. The comparison results in Figure 9 evidently show that Newton’s interpolation can effectively enhance the edge profile information of DoLP and AoP, and is very sensitive to the gradient information, but the method introduces artifacts, the distortion of polarized images, and is accompanied by certain meshing results. Bicubic, on the other hand, can guarantee the gradient information of polarized images, but the method introduces serious noise, which is particularly noticeable in AoP images. Compared to the above two traditional methods, ForkNet has some advantages in noise suppression, but the AoP images generated by this method have significant distortion. In conclusion, compared to the above methods, LLSPD-Net has better stability for polarization gradient characterization with less noise and higher contrast.

The quantitative comparison of the PDM models is shown in Table 2, which shows that LLSPD−Net is the best in all metrics except the PCQI metric of AoP. Moreover, the reason why the ForkNet contrast is high is because the method introduces a contrast constraint term. Although this method reduces noise, it also leads to the loss of polarization information. Therefore, overall, our LLSPD-Net has the best performance.

#### 4.2.3. Ablation Experiments

In order to verify the effect of each component in LLSPD−Net on the whole model, we performed ablation experiments. First, we removed the intensity network, as shown in Figure 10. It can be seen that without the enhancement of the low-illumination source image, the polarized image produced severe noise, which was caused by the lack of a high-quality RGB bootstrap image, resulting in artifacts in the complementary S_1,2_. In order to further explore the visual effect of the detail reconstruction module (DRM) in the intensity network for the enhancement of low-light images and the effect of polarization preservation, we removed the DRM module in the intensity network, as shown in Figure 10. Due to the module lacking in detail preservation and color correction, the enhanced RGB image produced color distortions. At the same time, from the solved Stokes vectors, i.e., the AoP and the DoLP images, it can be seen that the polarization information is seriously lost, and the polarization information generated is very far away from the GT, i.e., it generates polarization distortions. In addition, in order to explore the role of the repeat guidance (RG) module for the network in a polarization complementary network (PCN), we removed the module, and the results show that the generated polarized images are monotonically smoothed, which introduces less noise but loses the detailed information such as edges, gradients, etc., of the AoP and DoLP images.

The quantitative comparison of the ablation experiments is shown in Table 3, and it is clear that the LLSPD-Net combining the components of each part is the best in each index. It further proves that our method, using Stokes complementation, can significantly suppress the noise of polarized images. It also proves that our intensity network for low-illumination image enhancement does not cause any loss of polarization information, and that the polarization imaging is more accurate, at a high quality.

## 5. Conclusions

In this work, we propose a new sparse polarization imaging model for low-light environments based on Stokes vector completion. Specifically, we establish an incremental and synergistic polarization imaging model. This model contains two parts: the low-light image enhancement network and the Stokes complementary network. In addition, we acquired a paired polarization dataset (L-Polarization) that contains low- and constant-light field imaging for supervised network training. Both qualitative and quantitative results demonstrate that LLSPD-Net can enhance the realism and effectiveness of polarization image scene representation, and that our polarization imaging model for low-light environments is capable of acquiring high-contrast, high-fidelity polarization images as well as high-quality RGB images. In future work, we plan to design this model in a more concise manner to facilitate hardware piggybacking and implementation.

## Figures and Tables

**Figure 1 sensors-24-03299-f001:**
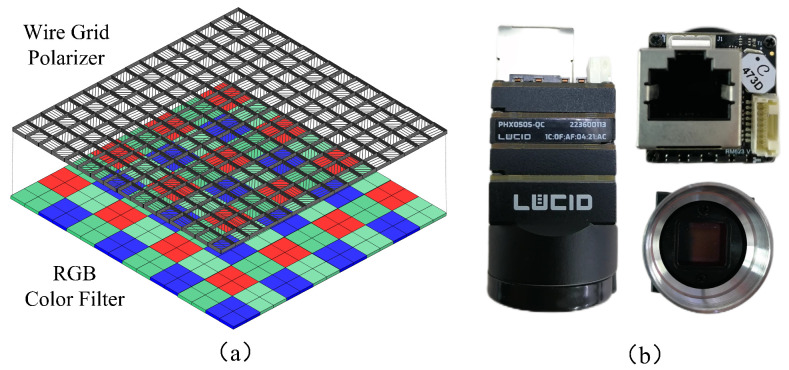
Polarization sensor structure. (**a**) An illustration of the polarization filter and Bayer filter distribution for the DoFP sensor. (**b**) The color polarization sensor used for the dataset acquisition in this paper.

**Figure 2 sensors-24-03299-f002:**
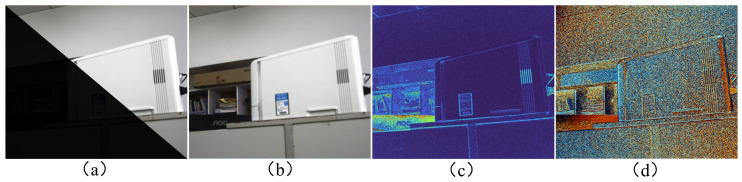
(**a**) The 0° image; the lower left is the low-light image, and the upper right is the image after direct luminance magnification. (**b**) The S0 image obtained from linearly amplified low-intensity images at 0°, 45°, 90°, and 135°. (**c**,**d**) The DoLP and AoP images, respectively, under low-light conditions.

**Figure 3 sensors-24-03299-f003:**
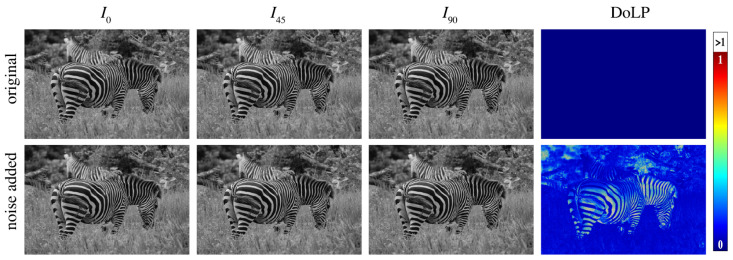
Noise creates polarization artifacts. Photo credit: Tim Caro.

**Figure 4 sensors-24-03299-f004:**
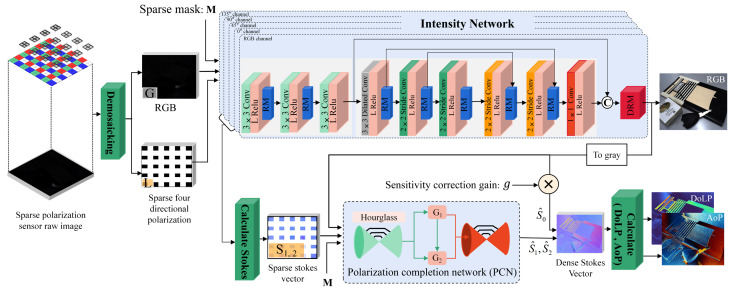
The inputs to this network are the RGB image of the raw image after demosaicing, the sparse polarization components S_1,2_, and the masked image M.

**Figure 5 sensors-24-03299-f005:**
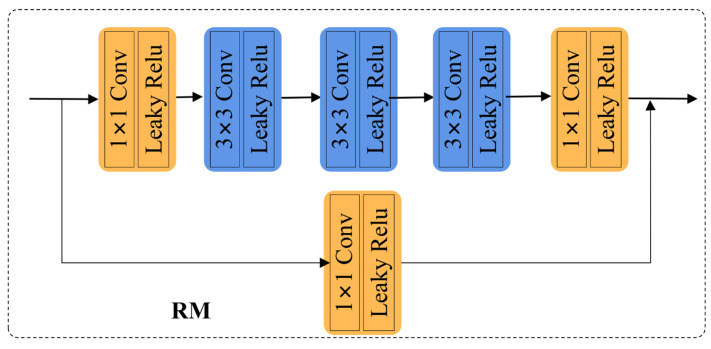
Specific structure diagram of the residual module (RM). It contains a total of 6 convolutional layers. One shortcut connection convolutional layer is included.

**Figure 6 sensors-24-03299-f006:**
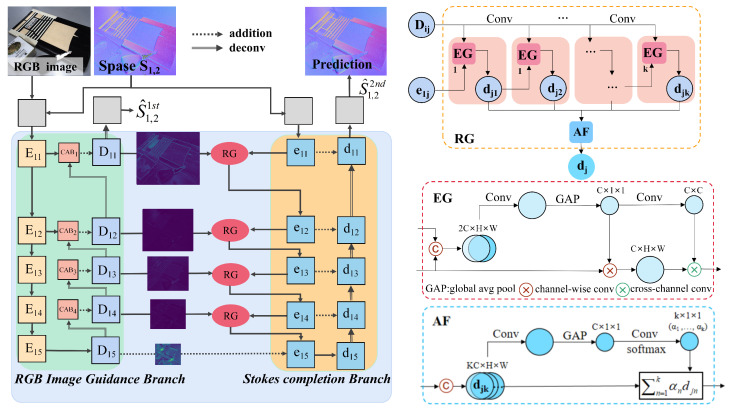
Polarization complementation network, which contains an RGB image guidance branch and a Stokes vector generation branch, where the repetitive guidance module RG is used to refine the Stokes Vector.

**Figure 7 sensors-24-03299-f007:**
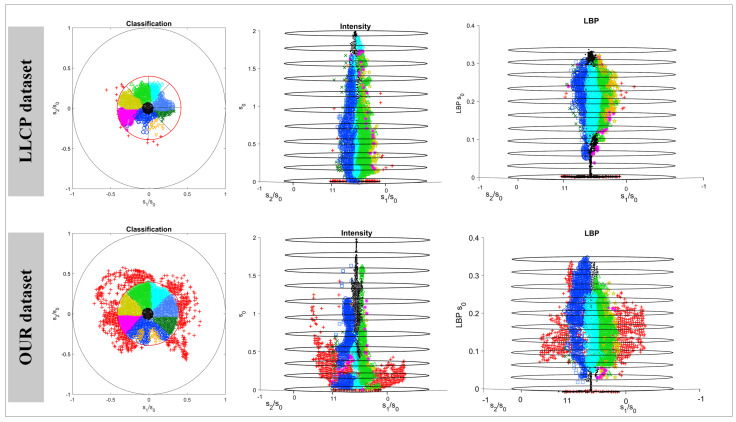
The dataset sample richness and completeness are represented in 2D and 3D forms by normalizing the Stokes parameter values of the polarization dataset. The first row is a categorical representation of the LLCP dataset samples. The second row is the categorical representation of the self−constructed L−Polarization dataset in this paper. The second column’s vertical coordinate indicates the median intensity value S0. The third column plot vertical coordinate indicates the median LBP value of S0.

**Figure 8 sensors-24-03299-f008:**
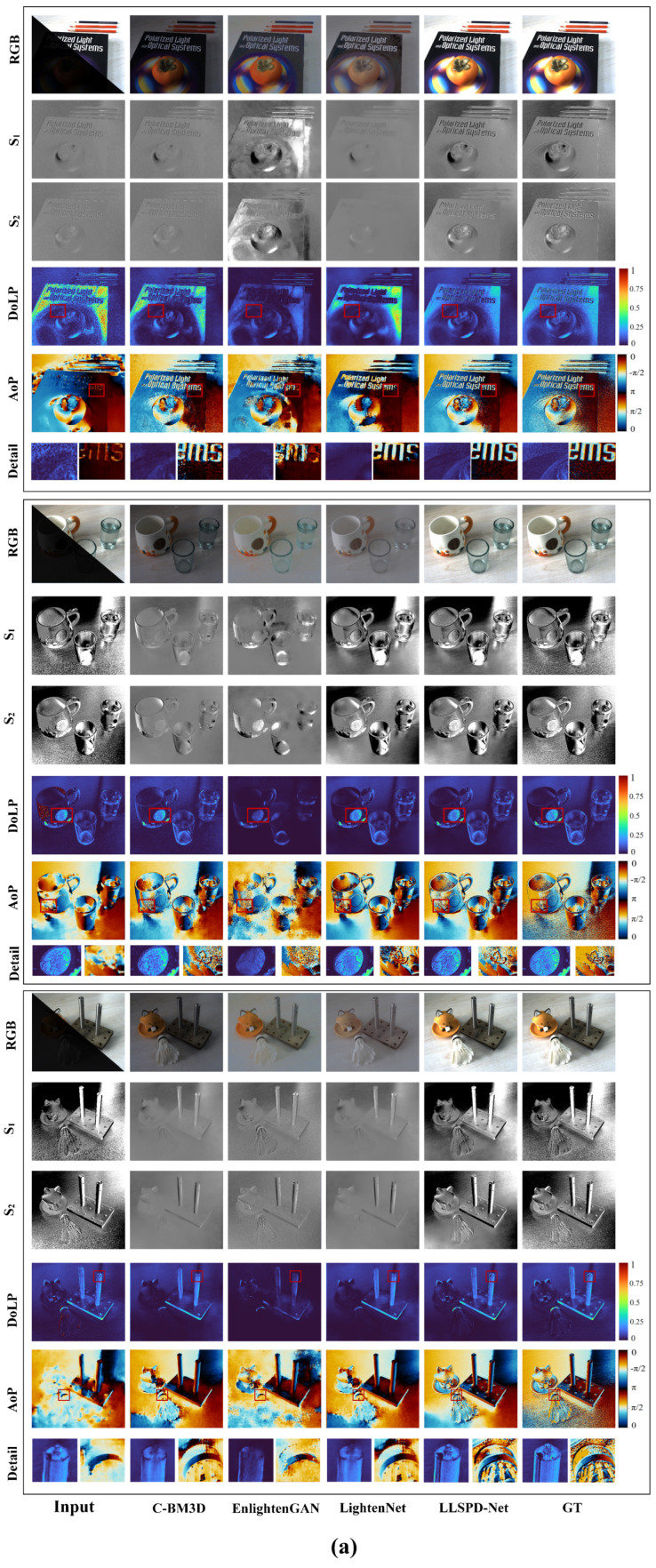
The results of RGB image enhancement, Stokes vector complementation, and DoLP and AoP imaging in low−light environment. (**a**) Indoor environment; (**b**) outdoor environment.

**Figure 9 sensors-24-03299-f009:**
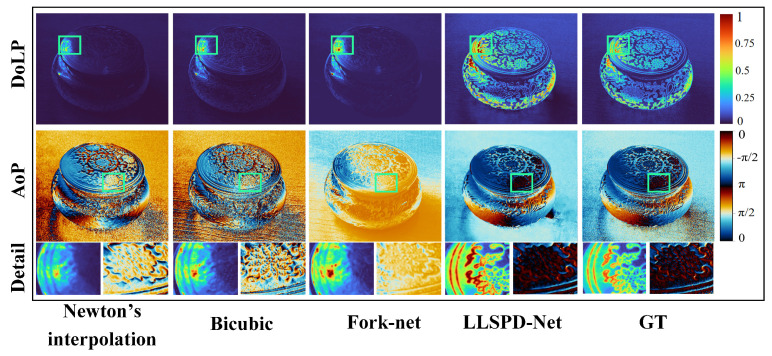
Results of qualitative comparison with several classical PDM models.

**Figure 10 sensors-24-03299-f010:**
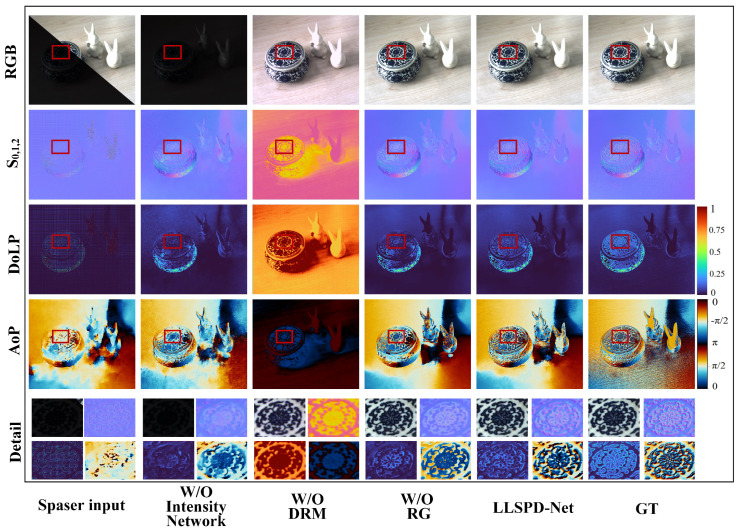
Ablation experimental validation of different components of the LLSPD−Net model. W/O Intensity Network denotes the result of direct polarization imaging without introducing a low-light image enhancement process, and W/O DRM denotes the effect of removing the detail enhancement module in the intensity network. W/O RG denotes the result of polarization imaging when removing the repetition guidance module in the Stokes completion network.

**Table 1 sensors-24-03299-t001:** Quantitative comparison for average PSNR, SSIM, PCQI, and angle error.

Method	RGB			S_1,2_			DoLP			AoP			
	PSNR	SSIM	PCQI	PSNR	SSIM	PCQI	PSNR	SSIM	PCQI	PSNR	SSIM	PCQI	Error [°]
Input	26.32	0.63	0.51	36.81	0.58	0.42	28.42	0.53	0.41	12.41	0.33	0.11	15.32
C-BM3D	28.59	0.81	0.83	38.26	0.79	0.69	30.16	0.57	0.46	12.52	0.26	0.37	19.51
EnlightenGAN	38.23	0.89	0.63	55.37	0.83	0.53	36.26	0.68	**0.69**	19.13	0.31	0.38	12.33
LightenNet	39.91	0.91	0.71	55.46	0.86	0.62	36.63	0.72	0.61	20.26	0.39	0.41	11.05
LLSPD-Net (ours)	**41.21**	**0.94**	**0.88**	**56.21**	**0.89**	**0.77**	**37.15**	**0.79**	**0.69**	**22.43**	**0.49**	**0.47**	**9.14**

**Table 2 sensors-24-03299-t002:** Quantitative Comparison for average PSNR, SSIM, and PCQI.

Method	DoLP			AoP			
	PSNR	SSIM	PCQI	PSNR	SSIM	PCQI	Error [°]
Newton’s	27.42	0.51	0.41	12.21	0.34	0.13	15.33
Bicubic	31.05	0.53	0.46	13.64	0.26	0.36	18.59
ForkNet	35.87	0.63	0.73	20.03	0.38	**0.45**	10.86
LLSPD-Net	**36.95**	**0.78**	**0.76**	**21.43**	**0.44**	0.42	**9.13**

**Table 3 sensors-24-03299-t003:** Ablation experiments for quantitative metrics comparisons.

Method	RGB			S1,2			DoLP			AoP			
	PSNR	SSIM	PCQI	PSNR	SSIM	PCQI	PSNR	SSIM	PCQI	PSNR	SSIM	PCQI	Error [°]
W/O Intensity Network	34.97	0.89	0.81	33.17	0.83	0.77	35.26	0.67	0.68	17.13	0.41	0.43	15.25
W/O DRM	31.47	0.75	0.76	31.41	0.78	0.79	31.39	0.58	0.61	18.46	0.42	0.39	20.23
W/O RG	35.68	0.94	0.85	34.38	0.85	0.81	35.57	0.69	0.73	22.87	0.47	0.45	11.73
LLSPD-Net	**36.84**	**0.96**	**0.87**	**35.47**	**0.88**	**0.84**	**36.87**	**0.74**	**0.76**	**23.85**	**0.51**	**0.47**	**10.15**

## Data Availability

The raw data supporting the conclusions of this article will be made available by the authors on request.

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
