# Peer review of "Low-Light Sparse Polarization Demosaicing Network (LLSPD-Net): Polarization Image Demosaicing Based on Stokes Vector Completion in Low-Light Environment"

_sensors, 2024, doi:10.3390/s24113299_

Round 1
Reviewer 1 Report
Comments and Suggestions for Authors
In this manuscript, the authors present a deep neural network architecture for demosaicing polarization RGB images in low-light environments. The network enhances low-intensity RGB images before passing the results into a polarization completion network to generate Stokes vectors. Comparative analysis is conducted against state-of-the-art deep learning-based methods and traditional interpolation-based techniques, demonstrating an improvement under low-light conditions. The manuscript holds promise and merits publication pending the resolution of the following concerns in the revised version:
1) The authors should specify the dominant noise source (e.g., read noise, Poisson noise) in the low-light conditions investigated, given their distinct statistical characteristics.
2) Please clarify the size of the training and testing datasets used in this study.
3) What is the minimum signal-to-noise ratio required for an input image to yield a meaningful enhanced result through the neural network?
Reviewer 2 Report
Comments and Suggestions for Authors
In the present paper, a model named low-light sparse polarization demosaicing network (LLSPD-Net) for simulating 7 sparse polarization sensor acquisition of polarization images in low-light environments has been proposed. The introduced model is theoretically feasible. Also, this research is supported by the convincing experimental results. The paper is well written, and the idea is innovative. Furthermore, this study is beneficial for researchers in the field of polarization imaging. Therefore, it can be accepted for publication after addressing the following minor revisions:
1. Please provide the source of Eq. (1).
2. Suggest splitting Figure 8 into two figures.
3. Table 2 shows that the model proposed in this article is not superior on all indicators. What is the reason for this? Can the authors continue to improve?
